# Do Mixed *Pinus yunnanensis* Plantations Improve Soil's Physicochemical Properties and Enzyme Activities?

Chen Liang [1], Ling Liu [2], Zhixiao Zhang [2], Sangzi Ze [3], Mei Ji [2], Zongbo Li [4], Jinde Yu [1], Bin Yang [4],* and Ning Zhao [1,4],*

1   College of Life Sciences, Southwest Forestry University, Kunming 650224, China; liangchenbb1310@163.com (C.L.); yujinde@swfu.edu.cn (J.Y.)
2   Yunnan Academy of Forestry and Grassland, Kunming 650201, China; liuling_km@126.com (L.L.); zzx_20071988@163.com (Z.Z.); meiji.emma@163.com (M.J.)
3   Yunnan Forestry and Grassland Pest Control and Quarantine Bureau, Kunming 650051, China; zesangzi@163.com
4   Key Laboratory of Forest Disaster Warning and Control of Yunnan Province, Southwest Forestry University, Kunming 650224, China; lizb@swfu.edu.cn
*   Correspondence: yangbin48053@swfu.edu.cn (B.Y.); lijiangzhn@swfu.edu.cn (N.Z.)

**Abstract:** Many survival and ecological problems have emerged in *Pinus yunnanensis* pure pine forest plantations that are usually assumed to be solved by creating mixed plantations. On this basis, we determined the physicochemical properties and enzyme activities of three soil layers in pure and three types of mixed *P. yunnanensis* plantation stands (admixed species: *Alnus nepalensis*, *Celtis tetrandra*, and *Quercus acutissima*) in Southwest China. We used one-way ANOVA with Tukey's test to analyze the effects of plantation type and depth on the soil's properties and variations among different depths. Principal component analysis combined with cluster analysis was used to evaluate the soil quality of different forest types comprehensively. The results showed that the stand with a mixing proportion of 2:1 of *P. yunnanensis* and *A. nepalensis*, *C. tetrandra*, and *Q. acutissima* had higher total porosity, moisture content, total nitrogen, available phosphorus, total phosphorus, sucrase, urease, and catalase enzyme activities than other proportions of mixed forest and *P. yunnanensis* pine pure forest. In general, the mixed *P. yunnanensis* plantation could improve the soil quality, especially its chemical properties and enzymes. This study provides a basis for creating a mixed-mode of *P. yunnanensis* and other tree species that can not only improve the economy of forest land but also enhance the ecological value.

**Keywords:** *Pinus yunnanensis* pure plantation; *Pinus yunnanensis* mixed plantation; soil; physico-chemical properties; enzyme activities; principal component analysis and cluster analysis

## 1. Introduction

*Pinus yunnanensis* is an ecologically and commercially important tree species distributed in subtropical Southwest China, with over 4 million hectares in plantations and natural forested areas [1]. The *P. yunnanensis* pine forests not only provide a large amount of high-quality timber and industrial raw materials but also play an important role in maintaining species diversity, conserving water resources, and retaining soil [2]. In recent years, because the composition of artificially constructed *P. yunnanensis* pure pine forests has become very simple and the planting area too large, a large number of survival and ecological problems in *P. yunnanensis* pine plantations have emerged. For example, diseases and insect pests are becoming more and more serious, the mountain ecosystem security has been seriously threatened, soil fertility and water retention have declined sharply, and biodiversity has been severely reduced [3,4]. Therefore, the outlook for pure coniferous forests is not optimistic. A change in the species composition and structure of forest stands, the biological characteristics, and the ecology of forest trees is needed to establish a more

complex and stable forest ecosystem than pure forests provide in order to increase the ecological and economic benefits of the forest. A large number of studies have proved that mixed forests play an important role in enhancing soil fertility, improving forest nutrient status, increasing forest species diversity, improving the stability of stand structure and productivity, and improving the ecological environment of forest plants [5–7]. Therefore, planting mixed forests is particularly important. Changing the species structure of forest stands leads to more complex and stable mixed forests [8]. Mixed forest is an important form of forest species and tree species allocation in the shelter forest system, and it is also a good way to form a reasonable structure of tree species. By selecting appropriate tree species to mix with *P. yunnanensis* pine, we can make full use of aboveground and underground energy and space so as to increase the biomass of mixed forests and improve the soil' nutrient status of forest land. This provides an important theoretical basis for the growth of forest vegetation and the construction of *P. yunnanensis* pine mixed forests.

Changes in plant diversity are known to affect aboveground and belowground ecosystem functioning, including the diversity of belowground communities of organisms [9–13]. Additionally, higher plant diversity may produce a higher biochemical diversity of root exudates, which further increases the soil's organic matter content [14]. Soil is an important part of the forest ecosystem and plays an important role in maintaining the stability of the terrestrial ecosystem, material circulation, and energy conversion [15]. Soil is also the foundation of plant growth and development. Soil fertility and quality directly determine the biological output and function of the forest. Foresters have always relied on knowledge of the soil's physiochemical properties to assess the value of the forests [16]. Soil fertility is a comprehensive reaction of the soil's physiochemical and biological properties [17]. In terms of chemical properties, the stoichiometric characteristics of soil N and P can reflect not only soil fertility but also the composition of soil's organic matter, soil quality, and the capacity of nutrient supplies [18]. Therefore, the content and distribution pattern of N and P available in the soil is important for plant growth. Soil enzymes can promote chemical reactions in organisms and are the total embodiment of soil's biological activity [19]. In addition, soil enzyme activities regulate the function of ecosystems and play a key role in nutrient cycling, which has been used by scholars as a parameter to evaluate soil quality [20]. The quality of the soil largely depends on the function of the soil, which indicates a combination of its physical, chemical, and biological characteristics [21]. The quality of soil nutrients is the result of the combined effects of soil physics, chemistry, biology, and other factors [22]. It is common to evaluate soil quality and nutrients by measuring the soil's physicochemical and biological indexes.

In recent years, a large number of experimental studies have focused on the innovation of *P. yunnanensis* cultivation technology, but they lack an understanding of the physical and chemical properties of *P. yunnanensis* plantation soil. Therefore, it is important to evaluate the advantages of different mixed forests by studying soil quality. In this study, the *P. yunnanensis* pine pure forest was used as the control group, and different mixed forest groups were set up to analyze and compare the physicochemical properties and enzyme activities of soil. This work provides a theoretical reference for establishing a mixed pattern of *P. yunnanensis* pine and other tree species, which could not only improve the forest economy but also enhance its ecological value.

## 2. Materials and Methods

### 2.1. Study Area Overview

The total management area of Dongshan Forest Farm in Dali city, Yunnan Province, China, is 98,300 hm$^2$, including 58,100 hm$^2$ of forest land (44,200 hm$^2$ of natural and 13,900 hm$^2$ of artificial forest). The topography of the forest farm is high in the east and low in the west. The highest altitude is 2046 m, the lowest altitude is 1416 m, and the average altitude is 1600 m. The annual sunshine in the forest farm is more than 2700 h, the frost-free period is 125–150 days, and the annual average precipitation is 500–550 mm. The soil type in this region is mainly neutral brown soil, with a small amount of cinnamon soil and mountain

meadow soil. In 2016, according to the distribution of typical forest types of the study area, six forest stands were constructed and set as study sites; I: *Pinus yunnanensis × Alnus nepalensis* (2:1); II: *Pinus yunnanensis × Alnus nepalensis* (3:1); III: *P. yunnanensis × Quercus acutissima* (2:1); IV: *P. yunnanensis × Quercus acutissima* (3:1); V: *P. yunnanensis × Celtis tetrandra* (2:1); VI: *P. yunnanensis* pure forest. These forest types belonged to undeveloped forest land before the construction of mixed forests, and the plants on the land were mainly *Heteropogon contortus* and *Dodonaea viscosa*. The six forest types are 100 m apart and do not interfere with each other. The tree species arrangement of the six forest types is shown in Figure 1. The soil's properties before the tree planting were very similar. We assumed that the local soil's properties were largely a consequence of plant growth and soil protection of forest types, and the initial conditions or management for the sites were similar. To reduce the effects of slope and elevation on the soil's properties, all the selected plots were taken at a similar slope (around 4) and elevation (around 1880 m) (Table 1).

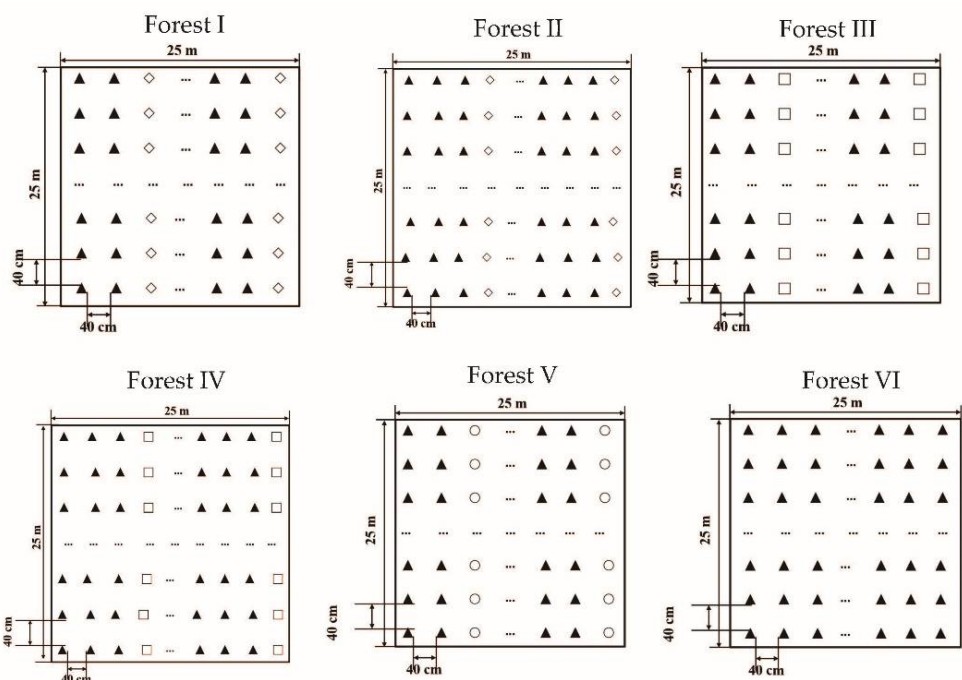

**Figure 1.** Area settings. Forest I: *Pinus yunnanensis × Alnus nepalensis* (2:1); Forest II: *Pinus yunnanensis × Alnus nepalensis* (3:1); Forest III: *Pinus yunnanensis × Quercus acutissima* (2:1); Forest IV: *Pinus yunnanensis × Quercus acutissima* (3:1); Forest V: *Pinus yunnanensis × Celtis tetrandra* (2:1); Forest VI: *Pinus yunnanensis* pure forest. ▲: *Pinus yunnanensis*; ◇: *Alnus nepalensis*; □: *Quercus acutissima*; ○: *Celtis tetrandra*, hereinafter the same.

**Table 1.** Basic information of experimental plots.

| Forest Stands | Altitude (m) | Slope (°) | Latitude | Longitude |
|---|---|---|---|---|
| I | 1890 | 3 | N 25°25′13.46″ | E 100°29′41.41″ |
| II | 1890 | 5 | N 25°25′15.17″ | E 100°29′39.50″ |
| III | 1879 | 5 | N 25°25′14.17″ | E 100°29′34.59″ |
| IV | 1895 | 2 | N 25°25′12.13″ | E 100°29′52.64″ |
| V | 1895 | 2 | N 25°25′12.13″ | E 100°29′52.64″ |
| VI | 1879 | 2 | N 25°24′56.76″ | E 100°29′36.00″ |

Notes: Forest I: *Pinus yunnanensis × Alnus nepalensis* (2:1); Forest II: *Pinus yunnanensis × Alnus nepalensis* (3:1); Forest III: *Pinus yunnanensis × Quercus acutissima* (2:1); Forest IV: *Pinus yunnanensis × Quercus acutissima* (3:1); Forest V: *Pinus yunnanensis × Celtis tetrandra* (2:1); Forest VI: *Pinus yunnanensis* pure forest.

### 2.2. Sample Plot Selection and Soil Sample Collection

In March 2016, we selected *P. yunnanensis* seedlings with consistent and healthy growth cultivated in seedling base to be transplanted to Dongshan Forest Farm in Midu County, Dali, Yunnan Province. The tree species used to set the same kind of mixed forest with different mixed proportions were also seedlings with consistent growth and no diseases and pests in the plantation. The soil samples were collected in mid-June 2016 and mid-June 2018 in this experiment.

Each plot of each forest type sample plot is more than 100 m apart, and the slope and planting management measures of the plantations are basically the same. In each forest type, three plots on the diagonal were taken, and the area of each plot is 625 m$^2$, i.e., 25 m × 25 m (Figure 2). Thus, a total of 18 standard plots were set up in our study sites. In a 625 m$^2$ plot, five large sample plots of size 5 m × 5 m were set in the four corners and center of each forest type plot to measure the ground diameter and tree height of *P. yunnanensis*. In each 625 m$^2$ plot, five small sample plots of size 2 m × 2 m were selected within the five large sample plots, and soil samples were divided into soil layers with a diameter of 4 cm. During sampling, litter on the soil surface was removed, and the sampling depth was 60 cm (divided into three layers of 0–20, 20–40, and 40–60 cm). Soil samples collected from five 2 m × 2 m plots of 625 m$^2$ were layered and mixed as one replicate, and a total of three 625 m$^2$ areas were used as three replicates. Samples were sealed in soil bags and taken back to the laboratory for testing. Each layer of soil samples in each repetition was divided into two parts. One part was sieved and stored in a refrigerator at 4 °C to determine the enzyme activities, and the other was dried naturally at room temperature to remove roots, stones, and other impurities in the soil sample and then screened for 2 mm to determine the physicochemical properties of the soil.

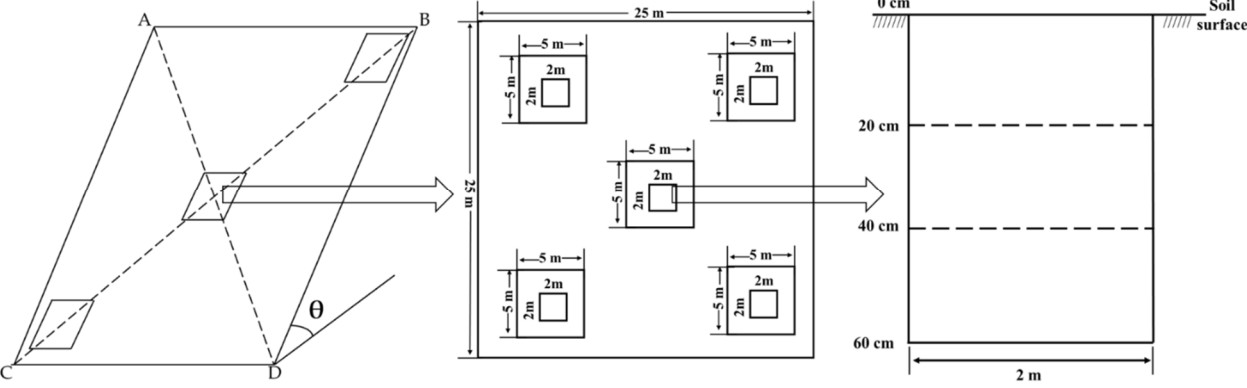

**Figure 2.** The diagram for the positions to collect soil samples in each plot.

### 2.3. Measurement Items and Methods

A set of three physical variables of the soil was evaluated: bulk density (BD), soil moisture content (MC), and total porosity (TOP). In the laboratory, undisturbed samples were weighed, dried in a forced-air oven at 105 °C for 48 h, and weighed again. The bulk density (BD, g/cm$^3$) and total porosity (T porosity, %) were determined by the cutting-ring method [23]. A total of 20 g of fresh soil was dried in an oven at 105 °C to a constant weight, and the soil's moisture content was measured by gravimetric analysis [24].

Four chemical variables of the soil were evaluated, pH, total nitrogen (TN), available phosphorus (AP), and total phosphorus content (TP). Soil pH was determined using a pH meter with a soil/water ratio of 1:2.5 [25]. The total nitrogen (TN) was determined by the automatic Kjeldahl method [26]. The total phosphorus (TP) was determined via the acid solution–molybdenum–antimony anti-colorimetric method [27]. The available phosphorus (AP) was extracted using 0.5 M NaHCO$_3$ (Olsen) and measured by a spectrophotometer [28].

Three enzyme activities of soil samples, which were stored in a refrigerator at 4 °C, were measured: urease activity (Ure), sucrase activity (Suc), and catalase activity (Cat). Soil urease activity (Ure) was determined using the phenol-sodium hypochlorite colorimetric method, sucrase activity (Suc) was determined using the 3,5-dinitrosalicylic acid colorimetric method, and catalase (Cat) activity was determined using the potassium permanganate titration method, respectively [29]. Urease activity and sucrase activity were expressed by the mass of specific substrate (or production of specific substrate) consumed by soil enzymes in 1 g of dried soil per unit time [30]. Catalase activity was expressed by the volume of 0.02 mol/L KMNO$_4$ consumed by soil enzymes in 1 g of dried soil per unit time [31].

We randomly selected ten *P. yunnanensis* pines from five 5 m × 5 m sample plots in each forest type and measured their ground diameter and plant height.

*2.4. Statistical Analysis*

At the beginning of afforestation (2016) and after two years of afforestation (2018), we selected 10 soil fertility indexes of six forest types for principal component analysis. The PCs receiving high eigenvalues and comprising variables with high factor loading were assumed to be the variables that best represent the system attributes. Therefore, we selected only PCs with eigenvalues $n > 1.0$. The score of each principal component is calculated according to the factor load of each principal component and the standardized mass fraction data. The calculation formula of principal component score (F) is:

$$F = FAC \times \lambda \tag{1}$$

where FAC is the normalized data and $\lambda$ is the arithmetic square root of the eigenvalue.

The composite principal component score (F $_{overall\ score}$) is the sum of the product of each principal component score and its corresponding contribution rate:

$$F_{overall\ score} = (F1 \times R1 + F1 \times R1 + + Fn \times Rn)/(R1 + R2 + \cdot Rn) \tag{2}$$

where R is the variance contribution rate.

The comprehensive score of each forest type was taken as a new index to evaluate its comprehensive quality, the difference of soil quality between different forest types was measured by Euclidean distance, and the classification average method and the shortest distance method were used to systematically cluster each treatment.

All statistical data analyses were performed using Microsoft Excel 2010 (Microsoft Redmond, WA, USA), Origin 8.0 software (Origin Lab Corp., Northampton, MA, USA), and SPSS software (Ver. 22.0; SPSS Inc., Chicago, IL, USA) for Windows. One-way analyses of variance (ANOVA) and Duncan's multiple comparison test were used to assess statistically significant differences ($p < 0.05$) among different soil depths under each forest stand. Pearson correlation analysis was conducted to identify relationships among measured the soil's properties.

## 3. Results

*3.1. Effects of Different Forest Types of P. yunnanensis on Soil's Physicochemical Properties*

3.1.1. Soil's Physical Properties

The results of three physical properties of soil under six forest stands are shown in Table 2. The soil bulk density showed an upward trend from the top of the soil that decreased with the increase in soil depth both at the beginning of afforestation (2016) and after two years of afforestation (2018). The interesting phenomenon is that the soil bulk density after two years of afforestation of the six forest types was all lower than that at the beginning of afforestation (2016); the soil bulk density of five *P. yunnanensis* mixed forest types was lower than that of the *P. yunnanensis* pure forest. However, total soil porosity and moisture content showed the opposite trend to BD; specifically, the total porosity and moisture content of six forest types showed a decreasing trend from the

surface layer as the soil depth increased. Of course, the total porosity and moisture content of the same layer after two years of afforestation (2018) were higher than those at the beginning of afforestation (2016). Besides this, compared with the *P. yunnanensis* pure forest, the *P. yunnanensis* × C. tetrandra (2:1) (Forest V) mixed forest stand in 0–20 cm soil layer had the lowest bulk density and the highest total porosity and water content after two years of afforestation (Figure S1). In addition, the soil's physical properties of different mixed proportions were also different in the unified mixed forest type. After two years of afforestation (2018), the total soil porosity and water content in Forest I and Forest III were all significantly higher than those in Forest II and Forest IV, respectively. However, at the beginning of afforestation (2016), there was no significant difference in the soil's physical properties among different mixed proportions of the same mixed forest type.

**Table 2.** Description of soil physical properties under six forest stands at 0–20, 20–40 cm, and 40–60 cm (mean ± stand error, $n = 3$).

| Year | Depth (cm) | I | II | III | IV | V | VI |
|---|---|---|---|---|---|---|---|
| | | | | BD (g/cm$^3$) | | | |
| 2016 | 0–20 | 1.29 ± 0.04 $^{Aa}$ | 1.26 ± 0.06 $^{Aa}$ | 1.25 ± 0.04 $^{Ba}$ | 1.26 ± 0.02 $^{Ba}$ | 1.29 ± 0.04 $^{Aa}$ | 1.28 ± 0.03 $^{Ba}$ |
| | 20–40 | 1.37 ± 0.05 $^{ABa}$ | 1.37 ± 0.07 $^{ABa}$ | 1.40 ± 0.08 $^{Aa}$ | 1.39 ± 0.02 $^{Aa}$ | 1.38 ± 0.07 $^{Aa}$ | 1.39 ± 0.04 $^{Aa}$ |
| | 40–60 | 1.4 ± 0.05 $^{Aa}$ | 1.41 ± 0.03 $^{Aa}$ | 1.42 ± 0.04 $^{Aa}$ | 1.42 ± 0.04 $^{Aa}$ | 1.42 ± 0.06 $^{Aa}$ | 1.41 ± 0.02 $^{Aa}$ |
| 2018 | 0–20 | 1.14 ± 0.07 $^{Aab}$ | 1.24 ± 0.04 $^{Aab}$ | 1.10 ± 0.02 $^{Aab}$ | 1.22 ± 0.02 $^{Aab}$ | 1.07 ± 0.15 $^{Ab}$ | 1.27 ± 0.03 $^{Ba}$ |
| | 20–40 | 1.19 ± 0.07 $^{Aab}$ | 1.28 ± 0.03 $^{Aab}$ | 1.20 ± 0.08 $^{ABab}$ | 1.31 ± 0.07 $^{Aab}$ | 1.13 ± 0.12 $^{Ab}$ | 1.36 ± 0.01 $^{Aa}$ |
| | 40–60 | 1.39 ± 0.18 $^{Aa}$ | 1.34 ± 0.05 $^{Aa}$ | 1.30 ± 0.06 $^{Aa}$ | 1.35 ± 0.07 $^{Aa}$ | 1.25 ± 0.05 $^{Aa}$ | 1.39 ± 0.04 $^{Aa}$ |
| | | | | TOP (%) | | | |
| 2016 | 0–20 | 44.53 ± 2.30 $^{Aa}$ | 46.88 ± 5.13 $^{Aa}$ | 46.86 ± 3.01 $^{Aa}$ | 45.88 ± 9.36 $^{Aa}$ | 45.36 ± 6.17 $^{Aa}$ | 44.61 ± 0.46 $^{Ca}$ |
| | 20–40 | 41.03 ± 1.18 $^{ABa}$ | 41.05 ± 7.02 $^{Aa}$ | 41.97 ± 2.00 $^{Aba}$ | 41.25 ± 4.76 $^{Aa}$ | 41.55 ± 1.89 $^{Aa}$ | 41.69 ± 1.47 $^{Aa}$ |
| | 40–60 | 37.51 ± 1.32 $^{Bb}$ | 37.65 ± 4.15 $^{Ab}$ | 37.43 ± 2.47 $^{Bb}$ | 37.21 ± 1.91 $^{Ab}$ | 37.21 ± 5.99 $^{Ab}$ | 37.75 ± 0.73 $^{Ba}$ |
| 2018 | 0–20 | 56.07 ± 0.57 $^{Aab}$ | 51.75 ± 1.61 $^{Abc}$ | 58.33 ± 4.50 $^{Aab}$ | 53.46 ± 3.08 $^{Abc}$ | 62.96 ± 2.79 $^{Aa}$ | 47.88 ± 1.46 $^{Ac}$ |
| | 20–40 | 52.89 ± 3.48 $^{ABab}$ | 48.83 ± 2.41 $^{ABbc}$ | 55.59 ± 1.84 $^{Aab}$ | 50.63 ± 1.87 $^{ABbc}$ | 59.2 ± 3.90 $^{Aba}$ | 43.61 ± 1.32 $^{Bc}$ |
| | 40–60 | 48.90 ± 0.70 $^{Bab}$ | 46.18 ± 1.04 $^{Bb}$ | 51.16 ± 1.92 $^{Aa}$ | 47.75 ± 1.38 $^{Bab}$ | 52.27 ± 2.71 $^{Ba}$ | 38.28 ± 1.76 $^{Cc}$ |
| | | | | MC (%) | | | |
| 2016 | 0–20 | 27.18 ± 1.01 $^{Aa}$ | 26.63 ± 6.58 $^{Aa}$ | 27.07 ± 11.31 $^{Aa}$ | 27.41 ± 4.49 $^{Aa}$ | 28.14 ± 6.36 $^{Aa}$ | 27.99 ± 1.20 $^{Aa}$ |
| | 20–40 | 22.32 ± 2.40 $^{Ba}$ | 22.1 ± 6.12 $^{Aa}$ | 23.84 ± 10.45 $^{Aa}$ | 22.07 ± 4.15 $^{Aa}$ | 22.78 ± 8.12 $^{Aa}$ | 22.72 ± 1.65 $^{Ba}$ |
| | 40–60 | 18.12 ± 0.24 $^{Ca}$ | 18.5 ± 4.94 $^{Aa}$ | 19.04 ± 4.45 $^{Aa}$ | 18.48 ± 5.35 $^{Aa}$ | 19.55 ± 6.60 $^{Aa}$ | 18.80 ± 1.33 $^{Ca}$ |
| 2018 | 0–20 | 38.09 ± 2.31 $^{Ba}$ | 34.69 ± 1.02 $^{Cb}$ | 39.44 ± 0.74 $^{Cab}$ | 33.22 ± 0.51 $^{Cb}$ | 42.05 ± 2.56 $^{Bab}$ | 28.90 ± 0.71 $^{Cc}$ |
| | 20–40 | 34.36 ± 1.19 $^{Aab}$ | 29.34 ± 0.74 $^{Abc}$ | 34.27 ± 2.55 $^{Aa}$ | 28.53 ± 1.04 $^{Ac}$ | 40.64 ± 2.82 $^{Aa}$ | 23.65 ± 0.70 $^{Ad}$ |
| | 40–60 | 29.36 ± 1.07 $^{Ab}$ | 25.52 ± 0.32 $^{Bc}$ | 27.56 ± 2.27 $^{Bb}$ | 24.16 ± 1.49 $^{Bc}$ | 27.77 ± 0.82 $^{Aa}$ | 19.65 ± 1.35 $^{Bd}$ |

Notes: BD: bulk density; TOP: total porosity; MC: moisture content. Different capital letters indicate that the indexes of different soil layers of the same forest type are significantly different ($p < 0.05$). Different lowercase letters indicate that there are significant differences in different indexes of different forest types in the same soil layer ($p < 0.05$).

### 3.1.2. Soil's Chemical Properties

Table 3 shows the results of four of the soil's chemical properties under six forest stands. Although the pH values of the three soil layers were not significantly different ($p > 0.05$), the soil of the six forest types were weakly acid, and its pH value was in the range of 4–7. Compared with the *P. yunnanensis* pure forest, *P. yunnanensis* mixed with *A. nepalensis*, *Q. acutissima*, *C. tetrandra*, and *P. yunnanensis* reduced the soil's acidity. The soil's TN, TP, and AP content from most of the forest stands significantly decreased with the deepening of the soil depth ($p < 0.05$). Interestingly, the TP and AP content after two years of afforestation (2018) were higher than those at the beginning of afforestation (2016). In addition, after two years of afforestation, Forest III had the highest mean TN content of both the soil depths, Forest V had the highest TP content of both the soil depths, and Forest I had the same AP content at both the soil depths (Figure S2). Besides this, the soil chemical

properties of different mixed proportions were also different in the unified mixed forest type. After two years of afforestation (2018), the soil's TN, TP, and AP content in Forest I and Forest III were all significantly higher than that in Forest II and Forest IV, respectively.

**Table 3.** Description of the soil's chemical properties under six forest stands at 0–20, 20–40, and 40–60 cm (mean ± stand error, $n = 3$).

| Year | Depth (cm) | I | II | III | IV | V | VI |
|---|---|---|---|---|---|---|---|
| | | pH | | | | | |
| | 0–20 | 4.71 ± 0.16 Ba | 4.74 ± 0.26 Aa | 4.74 ± 0.53 Ba | 4.74 ± 0.15 Aa | 4.69 ± 0.53 Aa | 4.72 ± 0.2 Aa |
| 2016 | 20–40 | 4.79 ± 0.02 ABa | 4.82 ± 0.19 Aa | 4.83 ± 0.52 ABa | 4.85 ± 0.40 Aa | 4.78 ± 0.69 Aa | 4.81 ± 0.18 Aa |
| | 40–60 | 4.92 ± 0.78 Aa | 4.91 ± 0.22 Aa | 4.96 ± 0.94 Aa | 4.92 ± 1.14 Aa | 4.94 ± 0.43 Aa | 4.93 ± 0.35 Aa |
| | 0–20 | 4.50 ± 0.72 Aa | 4.48 ± 0.51 Aa | 5.08 ± 0.34 Aa | 4.61 ± 0.27 Aa | 4.60 ± 0.57 Aa | 4.96 ± 0.32 Aa |
| 2018 | 20–40 | 5.30 ± 0.49 Aa | 4.79 ± 0.77 Aa | 5.05 ± 0.67 Aa | 4.68 ± 0.33 Aa | 4.75 ± 0.36 Aa | 4.83 ± 0.23 Aa |
| | 40–60 | 5.09 ± 0.72 Aa | 5.47 ± 0.38 Aa | 5.24 ± 0.35 Aa | 5.05 ± 0.16 Aa | 5.02 ± 0.21 Aa | 4.78 ± 0.32 Aa |
| | | TN (g/kg) | | | | | |
| | 0–20 | 0.46 ± 0.06 Aa | 0.51 ± 0.16 Aa | 0.50 ± 0.15 Aa | 0.47 ± 0.10 Aa | 0.47 ± 0.28 Aa | 0.48 ± 0.04 Aa |
| 2016 | 20–40 | 0.17 ± 0.02 Ba | 0.22 ± 0.08 Ba | 0.22 ± 0.09 Ba | 0.23 ± 0.01 Ba | 0.21 ± 0.03 Ba | 0.19 ± 0.04 Ba |
| | 40–60 | 0.15 ± 0.04 Ba | 0.17 ± 0.05 Ba | 0.16 ± 0.06 Ba | 0.18 ± 0.05 Ba | 0.13 ± 0.02 Ba | 0.14 ± 0.03 Ba |
| | 0–20 | 0.74 ± 0.03 Aab | 0.71 ± 0.04 Abc | 0.81 ± 0.04 Aa | 0.62 ± 0.02 Ac | 0.68 ± 0.05 Abc | 0.51 ± 0.04 Ad |
| 2018 | 20–40 | 0.49 ± 0.05 Ba | 0.29 ± 0.04 Bb | 0.56 ± 0.03 Ba | 0.26 ± 0.04 Bb | 0.32 ± 0.03 Bb | 0.22 ± 0.04 Bb |
| | 40–60 | 0.24 ± 0.03 Cab | 0.19 ± 0.04 Cab | 0.28 ± 0.04 Ca | 0.18 ± 0.03 Cb | 0.21 ± 0.03 Cab | 0.17 ± 0.04 Cb |
| | | TP (g/kg) | | | | | |
| | 0–20 | 0.24 ± 0.14 Aa | 0.23 ± 0.08 Aa | 0.27 ± 0.11 Aa | 0.23 ± 0.06 Aa | 0.27 ± 0.02 Aa | 0.25 ± 0.03 Aa |
| 2016 | 20–40 | 0.17 ± 0.1 Aa | 0.18 ± 0.05 ABa | 0.20 ± 0.07 Aa | 0.17 ± 0.05 ABa | 0.18 ± 0.06 Ba | 0.19 ± 0.04 Ba |
| | 40–60 | 0.15 ± 0.06 Aa | 0.14 ± 0.04 Ba | 0.14 ± 0.04 Aa | 0.14 ± 0.08 Ba | 0.15 ± 0.08 Ba | 0.16 ± 0.04 Ba |
| | 0–20 | 0.51 ± 0.34 Aab | 0.29 ± 0.06 Aab | 0.37 ± 0.05 Aab | 0.35 ± 0.04 Aab | 0.51 ± 0.23 Aa | 0.27 ± 0.02 Ab |
| 2018 | 20–40 | 0.29 ± 0.06 ABab | 0.24 ± 0.04 Aab | 0.31 ± 0.03 Aab | 0.27 ± 0.02 Bab | 0.34 ± 0.04 Ba | 0.21 ± 0.05 Ab |
| | 40–60 | 0.26 ± 0.03 Ba | 0.2 ± 0.08 Aa | 0.25 ± 0.03 Ba | 0.26 ± 0.03 Ba | 0.28 ± 0.02 Ca | 0.19 ± 0.06 Ba |
| | | Available P (mg/kg) | | | | | |
| | 0–20 | 2.29 ± 1.00 Aa | 2.31 ± 0.63 Aa | 2.32 ± 0.47 Aa | 2.31 ± 0.47 Aa | 2.24 ± 0.58 Aa | 2.27 ± 0.14 Aa |
| 2016 | 20–40 | 2.10 ± 0.37 Aa | 2.12 ± 0.24 ABa | 2.14 ± 0.75 Aa | 2.13 ± 0.22 Aa | 2.08 ± 0.54 Aa | 2.03 ± 0.06 Ba |
| | 40–60 | 1.91 ± 0.82 Aa | 1.93 ± 0.39 Ba | 1.94 ± 0.96 Aa | 1.95 ± 0.48 Aa | 1.84 ± 0.64 Aa | 1.87 ± 0.16 Ca |
| | 0–20 | 2.97 ± 0.05 Aa | 2.68 ± 0.28 Aab | 2.78 ± 0.24 Aab | 2.73 ± 0.15 Aab | 2.94 ± 0.16 Aa | 2.36 ± 0.11 Ab |
| 2018 | 20–40 | 2.53 ± 0.12 Ba | 2.27 ± 0.10 Bab | 2.45 ± 0.14 Bab | 2.25 ± 0.09 Bab | 2.32 ± 0.17 Bab | 2.14 ± 0.12 ABb |
| | 40–60 | 2.25 ± 0.21 Ca | 2.11 ± 0.07 Ba | 2.19 ± 0.14 Ba | 2.16 ± 0.03 Ba | 2.13 ± 0.30 Ba | 2.02 ± 0.21 Ba |

Notes: Available P: available phosphorus content; TP: total phosphorus content; TN: total nitrogen content. Different capital letters indicate that the indexes of different soil layers of the same forest type are significantly different ($p < 0.05$). Different lowercase letters indicate that there are significant differences in different indexes of different forest types in the same soil layer ($p < 0.05$).

### 3.2. Effect of Different Forest Types of P. yunnanensis on Soil Enzyme Activity

The results of soil enzyme activity in six different *P. yunnanensis* forest stands are shown in Table 4. The soil urease activity levels of *P. yunnanensis* forest types at two years of afforestation (2018) were all higher than those at the beginning of afforestation (2016), among which the urease activity of 0–20 cm surface soil was the highest. After two years of afforestation, the surface soil urease activity of Forest I was the highest, at 563.50 μg/g·24 h, which was significantly higher than that at the beginning of afforestation (2016) ($p < 0.01$) (Figure S3A). The soil sucrase activity of six different *P. yunnanensis* forest types all decreased significantly with the increase in soil depth ($p < 0.05$) both at the beginning of afforestation (2016) and after two years of afforestation (2018). The order of sucrase activity in different soil layers both at the beginning of afforestation (2016) and after two years of afforestation (2018) was 0–20 cm > 20–40 cm > 40–60 cm. As shown in Figure S3B, the sucrase activity of the surface soil after two years of afforestation (2018) of Forest

III was significantly higher than that of the surface soil at the beginning of afforestation (2016) ($p < 0.01$). The soil catalase activities of the six different forest types also showed a decreasing trend with the increase in soil depth both at the beginning of afforestation (2016) and after two years of afforestation (2018). After two years of afforestation (2018), the forest soil surface layer (0–20 cm) had the highest catalase activity, and Forest I had the highest mean catalase activity, which was 8.12 mg/g·24 h, and Forest VI had the lowest the catalase activity, which was 7.11 mg/g·24 h. As shown in Figure S3C, in the 0~20 cm soil layer after two years of afforestation (2018), there was no difference in the catalase activity of the soil under the same forest type. Obviously, the three enzyme activities of different mixed proportions were also different in the unified mixed forest type. After two years of afforestation, the three enzyme activities in Forest I and Forest III were all significantly higher than that in Forest II and Forest IV, respectively.

**Table 4.** Description of soil enzyme activities under six forest stands at 0–20, 20–40, and 40–60 cm (mean ± stand error, *n* = 3).

| Year | Depth (cm) | I | II | III | IV | V | VI |
|---|---|---|---|---|---|---|---|
| | | Ure (μg/g·24 h) | | | | | |
| 2016 | 0–20 | 268.61 ± 6.89 Ac | 345.85 ± 7.75 Aa | 229.45 ± 2.29 Ae | 226.32 ± 6.09 Ae | 248.72 ± 1.30 Ad | 294.97 ± 6.79 Ab |
| | 20–40 | 154.73 ± 4.61 Bcd | 196.69 ± 6.26 Ba | 178.14 ± 5.39 Bb | 199.66 ± 9.20 Ba | 144.16 ± 8.78 Bd | 167.48 ± 3.25 Bbc |
| | 40–60 | 127.03 ± 2.14 Ca | 95.00 ± 7.00 Cb | 121.93 ± 4.96 Ca | 116.7 ± 4.89 Ca | 115.06 ± 6.51 Ca | 72.12 ± 9.43 Cc |
| 2018 | 0–20 | 563.4 ± 12.56 Aa | 462.99 ± 5.06 Aab | 481.52 ± 1.27 Aa | 378.52 ± 11.33 Aab | 496.9 ± 5.39 Aab | 378.11 ± 7.04 Ab |
| | 20–40 | 447.34 ± 12.74 Ba | 346.08 ± 8.88 Bab | 329.24 ± 7.32 Bab | 263.08 ± 11.81 Bbc | 341.41 ± 5.65 Bb | 214.98 ± 4.18 Bc |
| | 31.93 | 125.54 ± 9.72 Ca | 122.84 ± 7.08 Bbc | 181.94 ± 5.26 Ca | 153.38 ± 4.92 Cbc | 154.58 ± 7.34 Cab | 148.88 ± 3.95 Bbc |
| | | Suc (mg/g·24 h) | | | | | |
| 2016 | 0–20 | 20.53 ± 1.62 Aa | 20.47 ± 2.78 Aa | 21.11 ± 1.57 Aa | 19.63 ± 2.35 Aa | 20.89 ± 3.57 Aa | 20.68 ± 2.36 Aa |
| | 20–40 | 13.15 ± 1.45 Ba | 14.17 ± 4.74 Aa | 12.94 ± 3.60 Ba | 14.04 ± 1.48 Ba | 13.91 ± 2.61 Ba | 13.27 ± 2.17 Ba |
| | 40–60 | 6.41 ± 1.40 Ba | 6.75 ± 0.57 Ba | 6.47 ± 1.83 Ca | 6.27 ± 0.61 Ca | 6.16 ± 0.87 Ca | 6.38 ± 0.58 Ca |
| 2018 | 0–20 | 28.58 ± 2.12 Bab | 24.79 ± 2.31 Abc | 33.23 ± 3.81 Aa | 27.91 ± 3.27 Aab | 31.93 ± 3.46 Aa | 22.59 ± 1.37 Ac |
| | 20–40 | 19.76 ± 3.58 Ba | 18.56 ± 11.67 ABa | 23.48 ± 2.78 Ba | 18.73 ± 1.99 Ba | 21.13 ± 3.36 Ba | 15.56 ± 1.99 Ba |
| | 40–60 | 9.16 ± 0.67 Cbc | 10.46 ± 1.04 Bbc | 13.56 ± 2.54 Ca | 8.97 ± 1.93 Cbc | 11.53 ± 1.92 Cab | 7.63 ± 0.58 Cc |
| | | Cat (mg/g·24 h) | | | | | |
| 2016 | 0–20 | 7.09 ± 1.06 Aa | 7.37 ± 0.45 Aa | 7.1 ± 0.51 Aa | 7.28 ± 0.77 Aa | 7.01 ± 0.58 Aa | 7.05 ± 0.14 Aa |
| | 20–40 | 6.51 ± 0.72 Aa | 6.87 ± 0.70 Ba | 6.46 ± 0.28 Aa | 6.81 ± 0.17 Aa | 6.45 ± 1.05 Aa | 6.49 ± 0.21 Ba |
| | 40–60 | 6.07 ± 0.78 Aa | 6.17 ± 0.48 Ba | 6.02 ± 1.04 Aa | 6.46 ± 0.47 Aa | 6.19 ± 1.55 Aa | 6.06 ± 0.15 Ca |
| 2018 | 0–20 | 8.12 ± 0.21 Aa | 7.48 ± 0.61 Aab | 7.52 ± 0.26 Aab | 7.36 ± 0.2 Aab | 7.74 ± 0.45 Aab | 7.11 ± 0.77 Ab |
| | 20–40 | 6.84 ± 0.64 Ba | 6.96 ± 0.77 Aa | 6.93 ± 0.52 ABa | 6.91 ± 0.69 Aa | 6.85 ± 0.51 Ba | 6.63 ± 0.18 Aa |
| | 40–60 | 6.65 ± 0.72 Ba | 6.23 ± 0.50 Aa | 6.56 ± 0.32 Ba | 6.54 ± 0.49 Aa | 6.48 ± 0.23 Ba | 6.18 ± 0.48 Ab |

Notes: Ure: urease activity; Suc: sucrase activity; Cat: catalase activity. Different capital letters indicate that the indexes of different soil layers of the same forest type are significantly different ($p < 0.05$). Different lowercase letters indicate that there are significant differences in different indexes of different forest types in the same soil layer ($p < 0.05$).

### 3.3. Principal Component Analysis and Cluster Analysis of Soil Fertility in Different Forest Types

As shown in Table 5, the first two components had eigenvalues of magnitude >1. The variance contribution rate of the principal component 1 (PC1) is 80.755%, which explains most of the variation in the data, the variance contribution rate of the principal component 2 (PC2) is only 11.770%, and the cumulative variance contribution rate of the PC1 and PC2 is 92.525%, which basically explains all the variation in the data. Taking the principal component scores of different forest types as a new index to evaluate the soil fertility quality, it can be seen that the soil fertility quality of different forest types is ranked as follows: 2018-V > 2018-I > 2018-III > 2018-IV > 2018-II > 2018-VI > 2016-II > 2016-III > 2016-IV > 2016-VI > 2016-V > 2016-I.

**Table 5.** Score and overall score of principal components influencing each index of soil fertility.

| Forest Type | FAC1 | F1 | FCA2 | F2 | $F_{overall\ score}$ | Ranking |
|---|---|---|---|---|---|---|
| 2018-I | 1.532 | 4.353 | −1.369 | −1.485 | 3.611 | 2 |
| 2018-II | 0.411 | 1.168 | −1.130 | −1.226 | 0.863 | 5 |
| 2018-III | 1.237 | 3.515 | 2.500 | 2.712 | 3.413 | 3 |
| 2018-IV | 0.384 | 1.091 | −0.359 | −0.389 | 0.903 | 4 |
| 2018-V | 1.667 | 4.736 | −0.183 | −0.198 | 4.108 | 1 |
| 2018-VI | −0.513 | −1.456 | 1.149 | 1.247 | −1.113 | 6 |
| 2016-I | −0.888 | 2.523 | −0.187 | −0.203 | −2.228 | 12 |
| 2016-II | −0.600 | −1.704 | −0.129 | −0.140 | −1.505 | 7 |
| 2016-III | −0.725 | −2.061 | 0.097 | 0.105 | −1.785 | 8 |
| 2016-IV | −0.809 | −2.298 | −0.104 | −0.113 | −2.020 | 9 |
| 2016-V | −0.864 | −2.455 | −0.192 | −0.208 | −2.169 | 11 |
| 2016-VI | −0.832 | −2.366 | −0.092 | −0.100 | −2.077 | 10 |

Taking Euclidean distance as a measure of the sudden fertility difference of different forest types, the results of the systematic clustering of each forest type by using the shortest distance method are shown in Figure 3. The twelve forest types were grouped into four categories, and 2018-I was separately grouped into one category; the soil quality was first class. 2018-II, 2018-III, and 2018-V were clustered into one class, and the soil quality was second class. 2018-IV, 2018-VI, and 2016-II were clustered into one class; the soil quality was third class. 2016-III, 2016-IV, 2016-V, 2016-I, and 2016-VI were clustered into one class; the soil quality was fourth class.

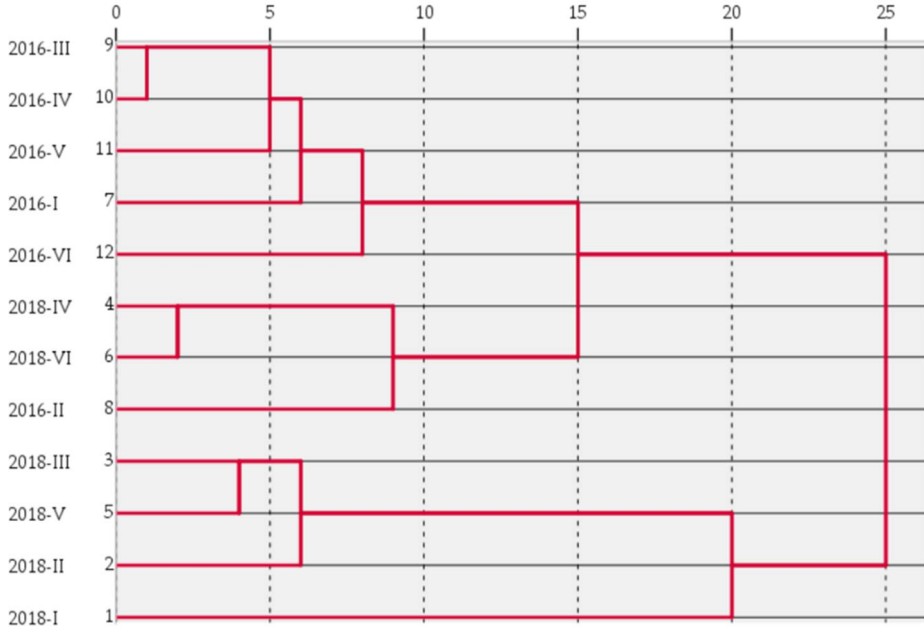

**Figure 3.** Hierarchical cluster analysis of the soil fertility of different forest types.

Pearson correlation analysis was used to examine the relationships among these properties to reduce redundancy (Table 6). Although the soil bulk density was not positively correlated with pH, it was significantly negatively correlated with other physical and chemical properties and enzyme activities ($p < 0.01$). This indicated that the larger the soil bulk density is, the worse the soil structure is, and the more compact soil is. This phenomenon will also lead to poor soil ventilation performance, which is not conducive to the decomposition and transformation of litter nutrients on the surface of forest land and thus will reduce the mass fraction of forest soil nutrients. There was a significant positive correlation between the soil water content and TN, AP, TP, and total porosity ($p < 0.01$). In

addition, the contents of total nitrogen, total phosphorus and available phosphorus, total porosity, and moisture content were significantly positively correlated with the activities of urease, invertase, and catalase ($p < 0.01$).

**Table 6.** Pearson correlation coefficients among soil properties.

|      | pH      | TN        | TP        | AP        | BD        | TOP      | MC       | Ure      | Suc      | Cat |
|------|---------|-----------|-----------|-----------|-----------|----------|----------|----------|----------|-----|
| pH   | 1       |           |           |           |           |          |          |          |          |     |
| TN   | −0.176  | 1         |           |           |           |          |          |          |          |     |
| TP   | −0.080  | 0.658 **  | 1         |           |           |          |          |          |          |     |
| AP   | −0.162  | 0.845 **  | 0.690 **  | 1         |           |          |          |          |          |     |
| BD   | 0.174   | −0.659 ** | −0.699 ** | −0.597 ** | 1         |          |          |          |          |     |
| TOP  | −0.086  | 0.624 **  | 0.723 **  | −0.670 ** | −0.704 ** | 1        |          |          |          |     |
| MC   | −0.094  | 0.773 **  | 0.755 **  | 0.727 **  | −0.763 ** | 0.881 ** | 1        |          |          |     |
| Ure  | −0.166  | 0.872 **  | 0.618 **  | 0.828 **  | −0.579 ** | 0.625 ** | 0.768 ** | 1        |          |     |
| Suc  | −0.163  | 0.859 **  | 0.690 **  | 0.745 **  | −0.681 ** | 0.642 ** | 0.775 ** | 0.814 ** | 1        |     |
| Cat  | −0.011  | 0.667 **  | 0.467 **  | 0.671 **  | −0.471 ** | 0.496 ** | 0.576 ** | 0.711 ** | 0.617 ** | 1   |

Notes: **, Significant correlation at $p < 0.01$. BD, soil bulk density; TOP, total porosity; MC: moisture content; TN, total nitrogen; TP, total phosphorus; AP, available phosphorus; Ure, urease; Cat, catalase; Suc, sucrase.

### 3.4. Growth Status of Various Tree Species after Two Years of Afforestation

By selecting the *P. yunnanensis* of four forest types and measuring its ground diameter and plant height, we found that the *P. yunnanensis* mixed with *A. nepalensis* or *Q. acutissima* has the highest ground diameter and plant height, which indicates that the *A. nepalensis* and *Q. acutissima* are beneficial to the growth of *P. yunnanensis* (Figure 4).

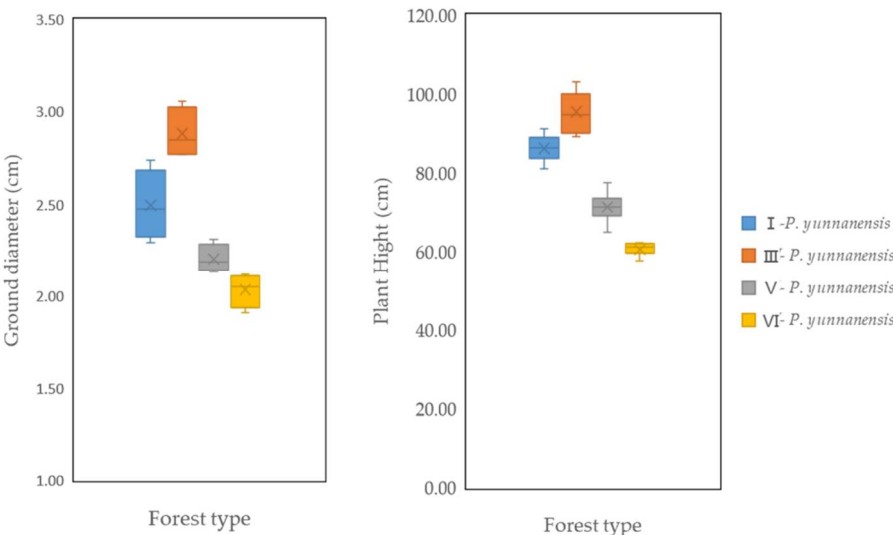

**Figure 4.** The growth status of *P. yunnanensis* of four forest types after two years of afforestation.

### 4. Discussion

Soil is the carrier of forest ecological processes and the material base of forest survival. Soil nutrient status directly affects the growth and metabolism of trees [32]. The mixing of different tree species inevitably leads to changes in the forest ecosystem, which directly affects the soil nutrient flow and material circulation, and the soil nutrient content also changes. Mixed forest stands are different, which leads to certain differences in different soil physical and chemical properties of different forest stands [33].

Li [15] and Lai [34] showed that different forest types have a significant impact on soil properties, such as soil's physical and chemical properties. Different forest types lead to certain differences in surface litter reserves and their composition, tree root growth and development, and the litter decomposition rate, resulting in different physical properties of

the soil of different forest stands [35]. The soil bulk density affects the circulation, storage, and distribution of water, moisture, and heat in trees. Small bulk density means loose soil texture and good structure. On the contrary, soil that is compact lacks a granular structure. This study found that the soil bulk density of the six forest types increased with increasing soil depth, which is consistent with previous studies [36,37]. Forest III and Forest V had the lowest bulk density, and Forest VI had the highest bulk density on the soil surface. This may be due to the single tree species of the *P. yunnanensis* pine pure forest, the slow decomposition of litter, and the soil becoming compact, which is not conducive to the long-term utilization of forest land. It indicated that *P. yunnanensis* pine mixed with other species had better soil infiltration. *Q. acutissima* is a broad-leaved deciduous tree species, the litter is easy to decompose, and the activity of the microbial community in the decomposition process can reduce the compactness of the soil. Soil moisture is an important component of the forest, which actively participates in the transformation and metabolism of soil substances [38]. In our study, the surface soil water content was higher than that of the deep layer, which may be because the litter layer of the mixed forest is thicker and the evaporation is less, so there is more surface soil moisture storage. The soil water holding capacity was different among different forest types, and the five mixed forests were higher than the *P. yunnanensis* pure pine forest, which was caused by the difference in soil porosity.

The changes in forest type, stand age, and land use can also affect soil's chemical properties significantly [39–42]. Generally, the litter of forest plants is distributed on the soil's surface, and a large number of nutrient elements are released on the surface of the soil. With the increase in soil depth, the litter is less and less [43,44]. Therefore, in this study, the total soil nitrogen, total phosphorus, and available phosphorus of each forest stand show the phenomenon of surface accumulation, which was consistent with the previous research [45]. After two years of afforestation (2018), the soil's TN, TP, and AP content in Forest I and Forest III were all higher than that in other forest types. The reason for this was that the nitrogen content in the middle leaf of *A. nepalensis* is higher than that in the leaves of other plants, and the degradation of leaf litter provides a large amount of nitrogen to the soil and improves soil fertility [46]. Forrester et al. [47] found that nitrogen content and nitrogen assimilation in the soil litter of mixed forests were higher than those of pure forests, which led to the content of available nitrogen and phosphorus in the soil of mixed forests being higher than that of pure forests. However, the soil's TN, TP, and AP content in Forest VI were lowest in six forest types; the reason is that the needles of *P. yunnanensis* have a coarse and hard texture, high cellulose content, waxy epidermis, and poor water permeability, so it is difficult to decompose and transform, which affects the accumulation of organic matter in soil [48]. This study also showed that the difference between TN and TP in mixed forests was basically not significant because soil TP mainly comes from rock weathering and litter decomposition, which is a long process [49]. TN mainly comes from the degradation of litter; the difference in TN among forest types is basically not significant, which may be related to the close litter and slow decomposition. In summary, the *P. yunnanensis* mixed forest had improved the soil quality, which was consistent with the research of Wang [50]. They showed that mixed forests promote increases in soil organic matter, N and P content, improve soil nutrient status, and help to sustain soil fertility.

In this study, after two years of afforestation, three enzymes of five *P. yunnanensis* pine mixed forest types were higher than *P. yunnanensis* pure pine. The results showed that the mixed forest was beneficial to the accumulation and decomposition rate of forest litter, increased the amount of nutrient return, improved nutrient availability, and accelerated soil nutrient mineralization [51]. Pure forest could inhibit the decomposition activities of soil microorganisms due to its allelopathic effect. Interestingly, the three kinds of soil enzyme activities in the surface soil (0–20 cm) of different mixed models were higher than those in the deep soil (40–60 cm). Yang et al. also found that the soil urease, sucrase, and catalase enzyme activities decreased with the increase in soil depth before and after afforestation, which was consistent with the research results of Yang [52]. The reasons for this vertical decrease in soil enzyme activity are that there were more litter and microorganisms in

the surface soil of the mixed forest, and the physiological and biochemical reactions of soil microorganisms and various enzymes were intense, while there were less litter and microorganisms in the deep soil, and the physiological and biochemical reactions were relatively stable [53]. Another reason is that the surface soil has a rich root system that penetrates the whole topsoil layer so that the enzyme activity in the surface soil is higher than that in the deep soil [54]. Sucrase comes from plant roots and microorganisms, which can catalyze the hydrolysis of sucrose into glucose and sucrose, which plays an important role in soil carbon and nitrogen cycling [55]. Catalase can promote the oxidation of various compounds by hydrogen peroxide, reduce the toxic effect of hydrogen peroxide in soil, and also reflect the total respiratory intensity in soil. In this study, different mixed modes had no significant effect on catalase activity, which may be related to tree species types and tree species growth stages. Zhou et al. [56] proved that the soil sucrase and catalase activities increased with the increase in forest age of the top Chinese prickly ash green food. In our study, the three enzyme activities in the topsoil of the six forest types after two years of afforestation (2018) were higher than those at the beginning of afforestation (2016). Therefore, long-term monitoring of soil enzyme activity in different mixed forests can be carried out in the future. Urease is a key enzyme in nitrogen cycling, which can catalyze the hydrolysis of urea. $NH_3$ formed by hydrolysis is one of the nitrogen sources of plants, and its activity can reflect the nitrogen supply capacity and level of soil [57]. In our study, after two years of afforestation, Forest I had the highest urease activities, which is mainly due to the presence of the nitrogen-fixing plant *A. nepalensis*. Nitrogen-fixing plants can often significantly increase soil organic matter and soil nitrogen [58]. For example, Wang et al. showed that the organic matter and nitrogen contents in the surface soil of an artificial forest of nitrogen-fixing trees were 40–50% and 20–50% higher than those of non-nitrogen-fixing trees, respectively [59]. At the beginning of afforestation (2016), there was no significant difference in soil indexes between mixed forest and pure forest. This may be because even though the forest types are different, this is a new mixed forest and their original soil conditions are basically the same.

Correlations were also found between these soil indices. There was a correlation between soil physicochemical properties and enzyme activities [60,61]. In this study, correlation analysis of physicochemical properties and enzyme activities of surface soil of six forest types was carried out. Soil enzymes are very sensitive to environmental changes, such as pH, temperature, and water content [62]. In our study, there was no significant negative correlation between soil pH and three kinds of soil enzyme activities, which indicated that if the soil's acidity is too strong, it will inhibit the enzyme activity. In addition, we selected ten soil fertility indexes of six forest types for principal component analysis to evaluate the soil quality of different forest types. Due to the different types of forest vegetation, there are some differences in the reserves and composition of surface litter, the growth and development of tree roots, and the decomposition rate of litter, resulting in the difference in the soil's nutrient content in different forest types [63]. In our study, through the comprehensive evaluation and ranking of soil quality of different *P. yunnanensis* pine mixed forest types by using the method of principal component analysis and cluster analysis, it was found that the soil quality of Forest V, Forest III, and Forest I were higher after two years of afforestation. The fundamental reason is that the species composition and biological characteristics of different forest types are different, so the quality, quantity, and decomposition rate of the litter of different forests are different, which affects the soil's nutrient content and distribution in different forest types. This indicated that the mixed pattern of *P. yunnanensis* pine growing for a certain number of years could improve the soil's nutrient content compared with the *P. yunnanensis* pure forest. On the other hand, compared with the growth status of *P. yunnanensis* under the *P. yunnanensis* pine pure forest, the growth status of *P. yunnanensis* pine under the three mixed patterns was relatively better, and the growth status of *P. yunnanensis* pine in Forest III was the best. This result was consistent with the result that the soil quality of Forest I, Forest II, and Forest V was better than that of Forest VI.



## 5. Conclusions

Interplanting different forests under *P. yunnanensis* forests can increase the soil quality, which is significant for improving the ecological environment of *P. yunnanensis* mixed forests. The results of this study showed that, compared with the *P. yunnanensis* pure forest, the *P. yunnanensis* × *A. nepalensis* mixed forest could improve soil physicochemical properties. The ratio of row spacing between *P. yunnanensis* and the selected mixed tree species was 2:1, which could make more rational use of soil nutrients and promote the growth of *P. yunnanensis*. Among them, such two planting patterns as *P. yunnanensis* × *Q. acutissima* mixed forest and *P. yunnanensis* × *A. nepalensis* have outstanding soil improvement effects, and they are worthy of promotion and application in *P. yunnanensis* forests in areas with insect pests.

Although the soil indexes of different *P. yunnanensis* mixed forests were measured in this study, their functions were not fully displayed because they were still in the young forest stage. Therefore, the future soil quality evaluation of the mixed forests needs a lot of research and long-term positioning observation. This study found that the mixed pattern of *P. yunnanensis* pine at the early stage of afforestation was affected by factors such as growth period and external environment, so proving its feasibility is a long-term process. Follow-up studies will continue in the later stage of the project to verify the effect of the *P. yunnanensis* pine mixed forest model and provide excellent companion trees for the sustainable management of *P. yunnanensis* pine mixed forests.

**Supplementary Materials:** The following supporting information can be downloaded at: https://www.mdpi.com/article/10.3390/d14030214/s1, Figure S1: Comparison of the physical properties of surface soil (0–20 cm) at the beginning of afforestation (2016) and after two years of afforestation (2018). Figure S2: Comparison of the chemical properties of topsoil (0–20 cm) in different forest types at the beginning of afforestation (2016) and after two years of afforestation (2018). Figure S3: Comparison of the enzyme activities in the surface soil (0–20 cm) of different forest types at the beginning of afforestation (2016) and after two years of afforestation (2018).

**Author Contributions:** Conceptualization, funding acquisition, and project administration, N.Z. and B.Y.; data curation. Z.L. and L.L.; formal analysis, C.L., Z.Z. and M.J.; investigation, S.Z. and M.J.; methodology, B.Y. and J.Y.; resources, C.L., S.Z. and N.Z.; software, L.L. and J.Y.; supervision, Z.Z. and Z.L.; validation, L.L. and S.Z.; visualization, C.L., B.Y. and J.Y.; writing—original draft, C.L.; writing—review & editing, C.L. and N.Z. All authors have read and agreed to the published version of the manuscript.

**Funding:** This work was supported by the National Natural Science Foundation of China (31760210) and the Key Project of Yunnan Applied Basic Research Program (Grant No.202101AS070009; 2018FG001-010).

**Institutional Review Board Statement:** Not applicable.

**Data Availability Statement:** Not applicable.

**Conflicts of Interest:** The authors declare no conflict of interest.

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
