# Peer review of "Do Mixed Pinus yunnanensis Plantations Improve Soil’s Physicochemical Properties and Enzyme Activities?"

_diversity, doi:10.3390/d14030214_

Round 1

Reviewer 1 Report

Comment on manuscript (No. 1573841) “Response of soil properties between pure and mixed Pinus yunnanensis plantations to tree species”

General comments

The authors of the manuscript studied soil property change in response to the mixed Pinus yannanensis plantations 2 year after tree planting.  There are comprehensive analysis of the data and results are worthy to be published.  However, the manuscript needs significant revision before publications.

Title: delete “to tree species”

Authors: Authors are missing in the list, no one was listed after “and”, and corresponding author is not on the list.

Abstract

Suggested editions are in the original manuscript.  Attention should be paid particularly to the unit. Those are very strange units. Available P usually expressed as mg P/kg soil, total P unit is % or g P /kg soil.  What is the unit g/kg3? The enzyme activity unit is also not seen. The enzyme activity unit is usually expressed in unit (mol, mmol, or umol substrate conversion per unit of time)

Page 1

L42, should be “are”, not “is”.

Page 2

For insect Tomicus yunnanensis, some place is written as T. yunnanensis in some place but in full name in other places.  Please be consistent.

L50-53. The description of impact of the T. yunnanensis is filled with empty words.  How it affects the bearing capacity, ecosystem degradation, environmental pollution.

L55. Please add “a” before “few”. Replace “chemical” with “synthesis”

L64-90. The paragraph is written like a textbook.  Only relevant publications should be cited.

L95-96. Select for a more diverse communities.  Who selects? What diverse communities?

Page 3

L100. “,” should be added after properties. Delete “so as”.  Replace “explore” with “determine”.

L101-103. Please delete sentence from Line 101 to Line 103.

L107. Delete “forest”.

L108. Is unit “hm2” used as a SI unit? Or Do English readers understand this unit? Location in latitude for the test site is needed.

L112. “(Chinese Soil Classification)” needs to be added. ~ is used in many places.  Please replace it with “-“.

L112-113. The sentence is not clearly written. 30-60 cm, depth of soil? More description of forest site grade I, II, and III are needed.

Page 4

All suggested changes are in the manuscript.

Page 5

L167. Brief description of enzyme is needed.

Also in the Material and Methods, measurement of tree size is missing.

Statistical analysis miss redundancy analysis.

Other suggestions are in the original manuscript.

Page 6

L223-224.  The sentence is not clearly written.

L236. In soil, aeration is generally used instead of ventilation.

Bulk density unit is not a SI unit.

Page 7.

The unit is not understood for total nitrogen in soil.

The authors need to explain why total N is not increased or changed but total P in the discussion.

Page 8

Please carefully check the units you used in the text and tables. 

L321. It is Fe3+ and Al3+, not Fe3+ and Al3+.

Page 9

Please check the unit for enzyme.

Other suggestions are in the manuscript.

Page 10

Units, please check, and make corrections.

Page 11

Please check and correct the units in Table 4.

Page 12

L413. When number is less than 10, please spell it out.  Should be “six” Instead of “6”.

Page 13

L438 lower case is needed for height.

Page 14

L457. Nitrogen in soil profile.  What do you mean?  A soil profile is from A horizon on the top all the way down to the bed rock.  More nitrogen in forest ecosystem should be on the upper horizons.

L488 “altitude gradient”, what is the meaning here? Vertical distribution?

L493-495. Sentence needs to be rewritten.

Page 15

Please be consistent with the insect’s name.

In the discussion section, there is no discussion on the redundancy analysis, why?

Supplementary materials are OK

Page 16 to 18.

Format of references are not consistent.  Some apparently is a copy and paste.

Details of the inconsistent are marked in the original manuscript.

Citation 20 (L606), last name and first name are switched. An unforgivable mistake for Chinese authors who cited publications from China.

Reviewer 2 Report

The study compares soil conditions in pure and mixed plantations of Pinus yunnanensis. Even though, I appreciate every study contrasting pure and mixed forests in a proper way, to broaden our general knowledge on the effects of tree species mixtures on different ecosystem functions, the study presented here has a lot of shortcomings. The scientific value is generally hard to judge as the study design and the actual size of plantations is not given. In the tables, a sample size of n = 3 is mentioned which is actually a very small sample size to draw any conclusions from. In addition, it is also an early stage after planting that was investigated. In the following, please find some more detailed comments:

  • The aim and scope of the study is, for example is not fully clear to me. You spent a whole paragraph talking about pests and specific pest species (Lines 44-63), but your study, comparing different young plantations regarding soil conditions, has nothing to do with pests. Changes in environmental condition may in the long run have an effect on the resistance against pests, but from 4-year-old plantations, no conclusion can be drawn on that. Therefore, also the last paragraph of your discussion is only speculative. Your data do not contribute to this. In the last sentence of the conclusion, you talk about Walnut forests. But this is not at all a topic of your paper. Or this information is largely missing in the whole manuscript. Maybe the farm you have worked on focuses on Walnuts, but this does not become clear from the paper.
  • Your study design is not properly presented. What is the size of you study plots? According to your Figure 1 you have strips that are planted with different tree species, but then this is no mixing or at least no mixed forests. How many plots do you have of each mixture type? In your tables you mention n = 3, but where are these three sampling plots. You do not exactly mention when the species were planted. You say that the mixtures were established in 2016, but what was there before. Was Pinus also only planted in 2016? What kind of plot areas have you then sampled in 2016, was this a clearcut site, that was freshly planted with different tree species combinations? Or was it the case that pine was already planted before and then other tree species were planted in between? But then the preconditions of the different forest types were not comparable, because the pines having been there earlier on the plots had more space to grow that those on the pure plots. It also does not become clear from your abstract that you investigated very young plantations only.
  • What is also missing is a short characterization of the natural vegetation there. Are mixed forests more resembling natural forests? Or are there only economical reasons for planting mixed stands?
  • In Figure 6 you present growth values of pine trees, but there is no mentioning of these measurements in the methods. How many trees were sampled? What was the initial size of the trees? Maybe there was a difference in height when planting.
  • Further, in the abstract you mention a TWO-Way Anova, but according to the methods section, you have done a one-way Anova. In general, n = 3 is a very low sampling number when contrasting soil conditions.
  • In the abstract you also refer to proper proportions in the end, but you do not go into details about the different mixing types and their potential differences. What was the reason then to investigate these different mixture types?
  • The results section is much too long and full of repetitions. It should be cut by half. The result numbers are already given in the tables, you do not have to mention them in the text again. Figure 2 to 4 are also redundant as you have the same information in the table. You also give wrong results in the text e. g. in line 258, when you say that the pH value after afforestation was higher than before. You did not find a significant difference for ph. The same is true for line 464, there was no significant difference in pH. There are other examples in the text where you talk about higher or lower values but there is no statistical significance.
  • Figure 5 has a very low quality and from the text it does not become clear what kind of sorting axis you mean. In line 399 you probably mean Figure 5b which shows the year 2018 (does not become clear what 5a and 5b figures show). Why using A, B, C… in Figure 5 and not the names (I, II, III IV etc.) used before. I also see no value when putting the years in two different figures. Put everything in one ordination figure and also display the direction of change for each plot.
  • I do not understand line 413 to 414. How can you deduce an inhibitory effect when there is no correlation between pH and enzyme activity?
  • You also already discuss some issues in the results part. This should be found in the discussion part e. g. line 428-435
  • Your discussion starts with insect resistance, but this is not what you have investigated.
  • You also focus a lot on the vertical decrease of certain variables such as enzyme activity, but this was observed in all forest types and is not the topic of your paper when you are interested in differences among forest types.
  • The paper needs language correction. There are many mistakes and sentences in the paper that do not make sense. In addition, species names are often not written in italics.

Reviewer 3 Report

This article aims to describe the effect of different types of afforestation with pine, in pure stand or in assocation with other species. The effect of sylviculture is particularly important, regards to all ecosystem services. In this article, the economical and environnemental aspects are likely well documented, relative to insects damages. Maybe, ecology could be more taken into account, in terms of interactions and consequences on the whole ecosystem biodiversity. However, since few articles present results about mix tree stands, this work is relevant and intersting.

Globally, the style should be improved for better understanding, all along the article. Some sentences could be cut or modified. The results shown in table are dense and difficult to read easily. I appreciate the figures.

Introduction

The introduction seems weel constructed and detailled to me. I am not able to evaluate the appropriateness of references. The objectives of the study are very clear.

Materials and Methods

I regret some lacking data concerning the site description. I don’t know this type of ecosystem and the given information didn’t allow me a good representation of the stand and the work.

  • I don’t know the functionning of your soil type. Maybe could you explain shortly what are the main constraints (I read about pH but others ?)
  • I would know more about the history of the stand : before afforestation, how was the stand : bare ground, forest, meadow? Was the cover cut, burned ?
  • How was the soil work for plantation?
  • How old are the trees at the begining of the experiment?
  • How long is a pine rotation in classical forestry? Or what represent 2 years regards to a forest rotation?
  • On the figure of your experimental design, I needed an idea of distances between trees, or of the soil blocks.
  • I regret not replicate of your blocks and the missing of a control bloxk without afforestation.

Concerning the protocols, I have some doubts. I am note sure that your map allow you to see the effect of differents mixtures. When I look at your map, the distance between two trees of on block is not largely more than the distance between two blocks. In other words, the influence of on tree will go wider than its block. We know that fungal hyphae spread very wide from one tree. In two years, mycorhizae can ever develop.

Your protocol for soil physical properties was not clear : I am not sure of the way you sampled soil, disturb and not (lines 147-149). How many replicates ?

You mention the effect of rhizodeposition. Maybe could you had carbon or organic matter analyses to your variables, as indicator of biological activity. We know and you cite the importance of microorganisms to the soil properties.

Why the ratio of 1:2.5 for pH (line 157). Is not 1:5 the norm ?

Why those enzymes ?

Results

Line 183-184 : effect of litter, could be in the discussion and more detailled : what type of litter, in quality, quantity ? What type of soil, with soil leaching, effect of clay…

I find your conclusions few robust, to see an effect of afforestation only by comparison of two years. I would prefer a comparison of pure and mixed stands to see a composition effect. Here you can’t be sure to distinguish the effect of the interannual variability and afforestation effect.

Discussion

At the beginning of your discussion, you mention the insects. We forget them during the results, and they land a little bit hard… If you want to keep this part, developp please !

You tested the effect after two years. I don’t know how old are your trees, and what it represents for a forest rotation. Maybe can you conclude for the early stage of afforestation, maybe not for a mature stand. The early stages may be critical for the durability of the stand, and vulnerability to insect damages. If it is, you could write or discuss it.

You describe the soil properties at three depths but your discussion is relatively poor in terms of functionnal aspects. How the mixing of species modify the soil activity, the understorey vegetation, and all microbial activities relative to rhizodeposition and litter decomposition.  Maybe could you had some consideration to root biomass, regards to the depth. It would help you for a better understanding of mecanisms implicated in what you observed.

Conclusion

Short conclusion, clear.

To conclude, you could improve the experimental design for more robustness. In your discussion, go further from the description to the functionning.

Round 2

Reviewer 2 Report

I thank the authors for their revision. The methods are now presented in a much more comprehensive way and the focus is more now on the comparison of soil conditions among mixture types. Still, you actually only have n = 1 for mixture type (the three soil samples are pseudoreplications), therefore, I am still not that convinced about the study. This limitation should at least be mentioned more explicitly.

The quality of presentation, however, does not justify publication yet. No language correction was done, species names are still not written in italics and there are still many sentences that do not make sense or are incomplete. Here just some examples:   L35-37; L44-45; L81-82; L127-128; L229; L262-263: L296-297

There is still no characterisation of the natural vegetation, at least I did not find it in the manuscript. How would the natural forests in this mountain region look like? What would be the dominating tree species?

Some other aspects that should be considered:

L23: When reading the abstract first, the reader has no idea what you mean with (2:1)

L28: You have not investigated the effect of mixed plantations on species diversity.

L35-37: Dou you want to say that high quality timber and industrial raw material are important for maintaining species diversity and water resources?

 L47-51: You start the sentence with “A large number of studies…”. At the of the sentence you only provide one reference.

L53-54: Do not understand what shelter forest systems have to with you young plantations.

L69: should it be the?

L104f: The admixed species were not mentioned before, therefore give the full name (and the names in italics)

Table 4: Explanation for this table is missing (lower case and capital letters…)

L349: Do not provide your personal feelings in a scientific paper. It is a scientific result. Maybe you low sample size also has something to do with finding no significant difference.

L358: at two years of

L390: Nitrogen-fixing plants

L412-413: You have not investigated an effect of underplanting on species diversity.

L427-428: I do not understand this statement as you have selected for similar environmental conditions before.

Reviewer 3 Report

First of all, I read your answer with attention and I thank you to have taken my questions into account. I particularly appreciate your emprovment of the Materials and Methods chapter, with a great effort for representing your experimental design.
However, this design is still lacking of robustness for me. Two points are particularly critical for me :
1. You sampled the soil for analyses on some subplots. It is a usual sampling design. After that, if I weel understand, you pooled all your samples for each treatment in one bag, and divided it into 3 technical replicates. I know that pooling can be useful for soil analyses, because of the large spatial variability at a fine scale. However, if you do that, you must ensure that you have real replicates for statiscal analyses. You can pool in a block for example, to « remove » the variability into the block, but but in that case, you have several blocks for your data table.
2. You don’t have kept any control without afforestation, to compare the soil properties before and after afforestation. You compare two different years, with no indication of an internnual effect.
For comments on each chapter, I will be short.
Introduction
I’m not sure that your changes in the introduction are gooing in the right direction. I found again the sentences with structure problems. You try to define the soil enzymes. A definition of enzyme is not useful, maybe you could put here your informations given in the discussion, and keep some confrontation with the littérature for the discussion.
Materials and Methods
Your chapter 2.4 should go into the 2.5. It is the description of a statistical analysis. You mention to test with oneway anova, only for the depth factor. In your results, you present in the table results of tests for forest types. Did you check for an interaction between those two factors ?
Results
Your tables are big !
Discussion
Your discussion is imporved regards to the previous version. However, you can go further with such an amount of results, discussing the interactions, correlations between variables, growth of trees regards to soil properties, or mechanistical approach of the properties that you measured.
To conclude, your paper was improved, but you can go further on this way. However, I’m not sure that the big problem of your experimental design can be upgraded.

Round 3

Reviewer 2 Report

Thank you for the revision. I recommend publication of this paper at the present form.

For further experiments you should consider at least two stands of each mixture type as you now only work with pseudpreplications and no real replications.

Reviewer 3 Report

I note a large work for a great improvment of the paper. Now, I find a large interest in publicating your results.